# Design and Analysis of Gallium Nitride-Based p-i-n Diode Structure for Betavoltaic Cell with Enhanced Output Power Density

**DOI:** 10.3390/mi11121100

**Published:** 2020-12-12

**Authors:** Young Jun Yoon, Jae Sang Lee, In Man Kang, Jung Hee Lee, Dong Seok Kim

**Affiliations:** 1Korea Multi-Purpose Accelerator Complex, Korea Atomic Energy Research Institute, Gyeongju 38180, Korea; yjyoon@kaeri.re.kr (Y.J.Y.); jslee8@kaeri.re.kr (J.S.L.); 2School of Electronic and Electrical Engineering, Kyungpook National University, Daegu 41566, Korea; imkang@ee.knu.ac.kr (I.M.K.); jlee@ee.knu.ac.kr (J.H.L.)

**Keywords:** betavoltaic cell, Gallium Nitride (GaN), high-output power, TCAD simulation

## Abstract

In this work, Gallium Nitride (GaN)-based p-i-n diodes were designed using a computer aided design (TCAD) simulator for realizing a betavoltaic (BV) cell with a high output power density (*P*_out_). The short-circuit current density (*J*_SC_) and open-circuit voltage (*V*_OC_) of the 17 keV electron-beam (e-beam)-irradiated diode were evaluated with the variations of design parameters, such as the height and doping concentration of the intrinsic GaN region (*H*_i-GaN_ and *D*_i-GaN_), which influenced the depletion width in the i-GaN region. A high *H*_i-GaN_ and a low *D*_i-GaN_ improved the *P*_out_ because of the enhancement of absorption and conversion efficiency. The device with the *H*_i-GaN_ of 700 nm and *D*_i-GaN_ of 1 × 10^16^ cm^−3^ exhibited the highest *P*_out_. In addition, the effects of native defects in the GaN material on the performances were investigated. While the reverse current characteristics were mainly unaffected by donor-like trap states like N vacancies, the Ga vacancies-induced acceptor-like traps significantly decreased the *J*_SC_ and *V*_OC_ due to an increase in recombination rate. As a result, the device with a high acceptor-like trap density dramatically degenerated the *P*_out_. Therefore, growth of the high quality i-GaN with low acceptor-like traps is important for an enhanced *P*_out_ in BV cell.

## 1. Introduction

Betavoltaic (BV) cells using a radioisotope have been developed for micro-battery applications, such as a power source of bio-medical implants and extreme environmental sensors [1,2,3], because of their long lifetime and micro-size. ^63^Ni radioisotope–based BV cells can be used for a long period due to a half-life of about 100 years. The BV cells based on various semiconductors such as Si [4,5], GaAs [6], SiC [7,8,9], GaN [10,11,12,13,14], and GaP [15] have been studied for high power conversion efficiency. Among the semiconductors, it is known that GaN-based BV cells can theoretically obtain superior conversion efficiency because of a wider energy bandgap. Moreover, GaN-based BV cells are more suitable for BV applications with long-term stability because GaN material has exhibited a strong radiation hardness [16,17], which can reduce the effects of radiation damage on device performances [18]. The p-i-n junction [10,11,12] and Schottky barrier diode [13,14] have been used to realize GaN-based BV cells. The p-i-n junction diode can obtain a wider depletion width, which improves the collection efficiency. However, the efficiency of the fabricated device exhibited a lower power conversion efficiency than the theoretical efficiency. Recent studies on BV cells aimed at enhancing the conversion efficiency. Many researchers have made progress in the optimization design of diode structures using a theoretical calculation model [19,20]. However, the design considerations of the GaN-based BV cells still have to be addressed due to the inherent properties of GaN material, including the various native trap states that occur during growth. The short-circuit current density (*J*_SC_) and open-circuit voltage (*V*_OC_) associated with the output power density of the diode are significantly affected by the native defects that influence recombination.

In this work, we optimized the GaN-based p-i-n diode to achieve a BV cell with high output power density using the following design parameters: heights of p-type GaN and intrinsic GaN (*H*_p-GaN_ and *H*_i-GaN_) and a doping concentration of i-GaN (*D*_i-GaN_). The physical phenomenon and performance were analyzed using a three-dimensional (3-D) technology computer-aided design (TCAD) simulator with physical models including e-beam irradiation and trap-assisted recombination models. The effects of native defects on *J*_SC_, *V*_OC_, and *P*_out_ of the e-beam-irradiated devices were also investigated.

## 2. Device Structure and Simulation Method

Figure 1 shows the 3-D schematic of the GaN-based p-i-n diode for BV cells. The p-i-n diode structure is the conventional diode structure, which consisted of an intrinsic GaN (i-GaN) region between p-type GaN (p-GaN) and n-type GaN (n-GaN) regions to obtain a wide width in the depletion region. The *D*_i-GaN_ determines the depletion width, which affects the conversion efficiency for BV cells. Here, i-GaN denotes undoped GaN, which is almost an n-type due to the residual donor [21]. The background impurity concentration in undoped GaN grown by metal-organic chemical-vapor deposition (MOCVD) is typically in the range of 10^15^ to 10^17^ cm^−3^, depending on the condition of the reactor [22]. Furthermore, it is difficult to grow undoped GaN with simultaneous low doping concentration and high quality [23]. Thus, the *D*_i-GaN_ was varied in the range of 5 × 10^15^ cm^−3^ to 5 × 10^16^ cm^−3^ for optimizing the *D*_i-GaN_. In order to reduce the resistance of the n-GaN and p-GaN layers, doping concentrations of the n-GaN and p-GaN (*D*_n-GaN_ and *D*_p-GaN_) were designed as 5 × 10^18^ cm^−3^ and 5 × 10^17^ cm^−3^, respectively. We also changed the *H*_p-GaN_ and *H*_i-GaN_ to achieve high performance. The variation ranges of *H*_p-GaN_ and *H*_i-GaN_ were 60–200 nm and 500–900 nm, respectively. The ranges of *H*_p-GaN_ and *H*_i-GaN_ values were determined by considering the penetration depth of 17 keV electrons at about 1 μm [24]. The energy of the e-beam is the average energy of ^63^Ni [25,26]. The contact resistance for p-GaN and n-GaN in the devices was 1 × 10^−4^ Ω·cm^2^ [27,28].

The effects of e-beam irradiation on current characteristics were analyzed using a TCAD simulator [29]. Physical models were applied in the simulation, including Schockley–Read–Hall (SRH) and trap-assisted tunneling (TAT), and a low-field mobility model. The SRH and TAT models were used to reflect the carrier recombination phenomenon [30], which significantly affects the *J*_SC_ and *V*_OC_ of the diode. For the effects of native defects in the GaN material [31,32], acceptor and donor-like trap states were added to the simulation. The acceptor-like trap state is formed by Ga vacancies. The donor-like trap states are mainly associated with the N vacancies and nitrogen antisite point defect. When we optimized the structure, we analyzed the performances of the diodes applied by native traps. In addition, the individual impact of trap states on *P*_out_ were studied to investigate the dominant traps that degrade the performances.

## 3. Results and Discussion

Figure 2a shows the effects of e-beam irradiation on the reverse current characteristics of the GaN p-i-n diode. When the diode was irradiated by a 17 keV e-beam, the reverse current density significantly increased. This was because electron-hole pairs (EHPs) were generated by the injected high-energy electrons. The electrons and holes in the depletion region were respectively moved by internal electric field through n-GaN and p-GaN regions and the carriers converted to the electric current. The *J*_SC_ and a *V*_OC_ of the irradiated diode were 14.92 μA/cm^2^ and 2.391 V, respectively. The *J*_SC_ and *V*_OC_ were defined as the current density at a voltage of 0 V and the voltage when the current density was 0 A/cm^2^, respectively.

As shown in Figure 2b, the 17 keV electrons penetrated up to a depth of about 1 μm and the peak absorption rate was exhibited at a depth of about 300 nm. Because many EHPs generated in the i-GaN region contributed to the conversion efficiency, the *H*_i-GaN_ and *D*_i-GaN_ are important design parameters. A high conversion efficiency increases *J*_SC_ and *V*_OC_, which influences the output power density (*P*_out_). The diode exhibited the maximum *P*_out_ (*P*_out_max_) at a voltage of 2.18 V, as shown in Figure 2c.

Figure 3a shows the variations of the reverse current characteristics of the irradiation diodes as a function of *H*_i-GaN_. As the *H*_i-GaN_ increased, the reverse current density became higher due to extension of the absorption region. Many EHPs were generated in the extended absorption region, which converted the electric current. However, the current density of the diodes with a *H*_i-GaN_ above 900 nm was lower than that of the device with a *H*_i-GaN_ of 700 nm at a forward voltage above 0 V. This result indicated that excess carriers generated by the e-beam were reduced by the recombination mechanism as they moved through the n-GaN or p-GaN regions. This result affected the *V*_OC_ and *J*_SC_ of the diodes. As shown in Figure 3b, the device with a *H*_i-GaN_ of 700 nm had the highest *V*_OC_. In terms of *P*_out_max_, the device with a *H*_i-GaN_ of 700 nm exhibited the highest *P*_out_max_ because the device was less affected by the recombination phenomenon, as shown in Figure 3c.

The variations of *J*_SC_, *V*_OC_, and *P*_out_max_ depending on the *H*_i-GaN_ and *H*_p-GaN_ are shown in Figure 4. The *J*_SC_ of the devices increased with a rise in the *H*_i-GaN_ regardless of the *H*_p-GaN_ because a high *H*_i-GaN_ extended the absorption region. When the *H*_i-GaN_ above 800 nm increased, the *J*_SC_ slightly decreased due to the recombination phenomenon. As the *H*_p-GaN_ became shorter, the variation of *J*_SC_ according to the *H*_i-GaN_ increased. This result indicated that the absorption rate vs. depth was more affected in the irradiated device with a short *H*_p-GaN_. The device with a *H*_p-GaN_ of 60 nm and *H*_i-GaN_ of 800 nm was the highest *J*_SC_ because the *J*_SC_ was enhanced by an additional absorption near the p-GaN region. In case of the *V*_OC_, the device with a *H*_p-GaN_ of 100 nm and a *H*_i-GaN_ of 700 nm exhibited the highest *V*_OC_, as shown in Figure 4b. Because a short *H*_p-GaN_ degenerated the carrier transport, the *V*_OC_ of the devices with a *H*_p-GaN_ of 60 nm was a smaller than that of the devices with a *H*_p-GaN_ of 100 nm. The device also obtained the highest *P*_out_max_. The *P*_out_max_ value was affected by a change of the *V*_OC_.

Figure 5 shows energy band diagrams of the diodes with different values of *D*_i-GaN_. As the *D*_i-GaN_ decreased, the depletion region was extended in the i-GaN region. Because the depletion region was influenced by the *D*_i-GaN_, we examined the variations of *J*_SC_, *V*_OC_ and *P*_out_max_ depending on the *D*_i-GaN_ and *H*_i-GaN_. As shown in Figure 6, the diode with a low *D*_i-GaN_ exhibited an improved *J*_SC_ because of a wider depletion region in the i-GaN region. The excess carriers could be moved by the built-in electric field. Although the device with a *D*_i-GaN_ of 1 × 10^16^ cm^−3^ exhibited a higher *V*_OC_, the diode with a *D*_i-GaN_ of 5 × 10^15^ cm^−3^ obtained a higher *P*_out_max_. This result indicated that reducing the *D*_i-GaN_ is important to improving transport efficiency. However, in terms of GaN epitaxial technology based on MOCVD method, it was difficult to reduce the *D*_i-GaN_ below 1 × 10^16^ cm^−3^ because residual impurities remained during the growth process. Therefore, we determined that the optimum point for the *D*_i-GaN_ was 1 × 10^16^ cm^−3^. As a result, the diode structure with a *H*_p-GaN_ of 100 nm, *H*_i-GaN_ of 700 nm, and *D*_i-GaN_ of 1 × 10^16^ cm^−3^ was optimized, and the effects of native trap states on performances of the optimized diode were investigated with variations of trap level and density.

The energy spectrum of the ^63^Ni source exhibited a wide range to a 66 keV peak energy [19]. We additionally confirmed the performances of the optimized diode depending on the injected electron energy. The current characteristics of the diodes irradiated by different e-beam energies is shown in Figure 7a. As the energy increased up to 30 keV, the current became higher. This was because many EHPs were generated by a wide distribution of absorption rate as shown in Figure 7b. However, when the electrons with an energy above 40 keV were injected, the current of the irradiated device was lower than that of the device irradiated by the 17 keV e-beam. These results revealed that the variations of the current of the irradiated diodes depending on the energy of the e-beam were large. The probability of beta particles generated from the ^63^Ni source showed a high distribution below 20 keV [19]. Also, the depletion width formed in the i-GaN region was small at about 600 nm (in case of *D*_i-GaN_ = 1 × 10^16^ cm^−3^), and the diffusion length of GaN can be shortened by native defects. Therefore, in order to achieve a high efficiency BV cell using the ^63^Ni source, it is necessary to analyze the performances of GaN-based diodes considering the spectrum of the ^63^Ni source.

Figure 8 shows reverse current density and *P*_out_ of the irradiated diodes with and without the native trap states including donor and acceptor-like traps. The reverse current density was significantly degenerated by the trap states. This result means that the trap-assisted recombination was caused by the native defects in the GaN material. As a result, the *P*_out_ of the device with the trap states was lower than that of the device without the trap states. We confirmed the effects of individual traps on current characteristics of the irradiated devices. As a shown in Figure 9, the impact of acceptor-like trap states was stronger than that of donor-like trap states. This result proved that the acceptor-like trap states represented the dominant factor for the recombination.

In addition, we investigated the effects of trap density on *J*_SC_, *V*_OC_, and *P*_out_max_. As the trap density increased, the performances were totally degenerated by the acceptor-like trap state, as shown in Figure 10. The donor-like trap states reduced the *J*_SC_ less than the acceptor-like trap state. While the donor-like trap states (*E*_C_-0.6 eV) slightly decreased the *V*_OC_, the shallow donor-like trap states (*E*_C_-0.23 eV) increased *V*_OC_. As a result, the *P*_out_max_ was unaffected by the shallow donor-like trap states (*E*_C_-0.23 eV). It is important to reduce the acceptor-like traps to improve the conversion efficiency of the betavoltaic cell.

## 4. Conclusions

In this work, we designed a p-i-n diode with a variation of geometric parameters, namely, *H*_i-GaN_, *D*_i-GaN_, and *H*_p-GaN_, and analyzed *P*_out_ using 17 keV e-beam irradiation. The *H*_i-GaN_ and *H*_p-GaN_ affected the absorption rate vs. depth. A low *D*_i-GaN_ produced an increase in depletion width. The optimized structure with a *H*_i-GaN_ of 700 nm, *D*_i-GaN_ of 1 × 10^16^ cm^−3^, and *H*_p-GaN_ of 100 nm obtained an improved *P*_out_. In addition, the effects of native trap states on reverse current characteristics were investigated with various trap levels and densities. When the acceptor-like trap density increased from 10^14^ cm^−3^ to 10^16^ cm^−3^, the trap significantly decreased the *P*_out_max_ by about 15%. GaN with low acceptor-like traps was needed to enhance the *P*_out_ of a BV cell. These results provide design considerations for achieving a high efficiency BV cell.

## Figures and Tables

**Figure 1 micromachines-11-01100-f001:**
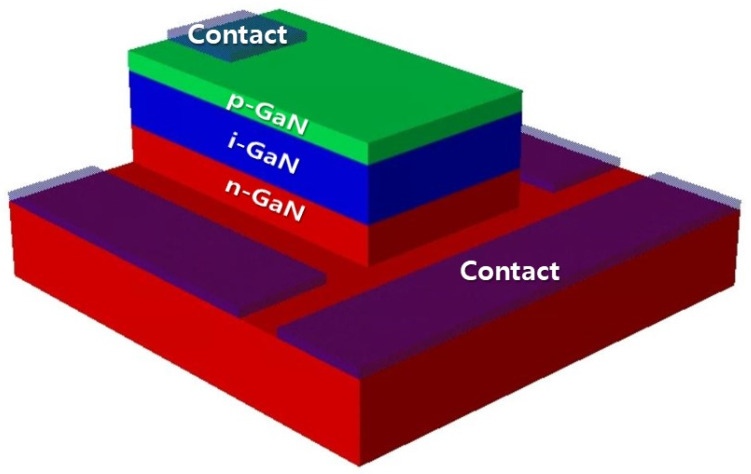
3-D schematic of the p-i-n diode structure for betavoltaic (BV) cell.

**Figure 2 micromachines-11-01100-f002:**
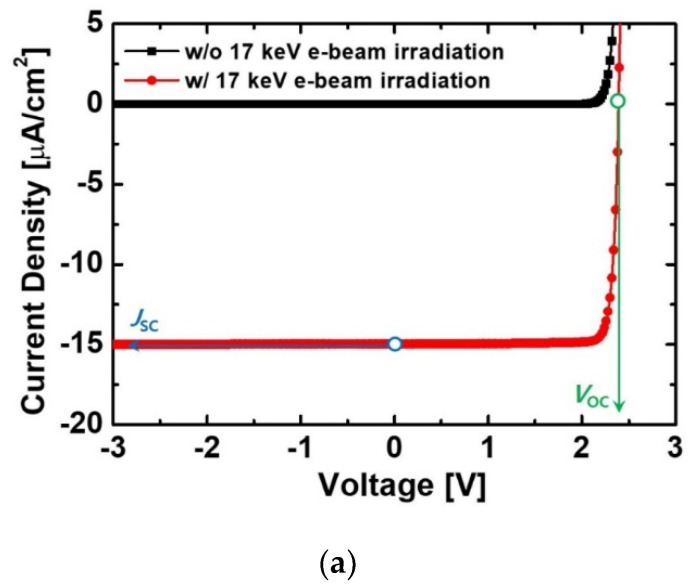
(**a**) Effects of e-beam irradiation on reverse current characteristics of the diode. (**b**) Absorption rate vs. penetration depth in the diode for 17 keV e-beam irradiation. (**c**) The output power density of the e-beam-irradiated diode. *H*_p-GaN_ and *H*_i-GaN_ in the diode were 100 nm and 600 nm, respectively. *D*_p-GaN_, *D*_i-GaN_, and *D*_n-GaN_ were 5 × 10^17^ cm^−3^, 1 × 10^16^ cm^−3^, and 5 × 10^18^ cm^−3^, respectively.

**Figure 3 micromachines-11-01100-f003:**
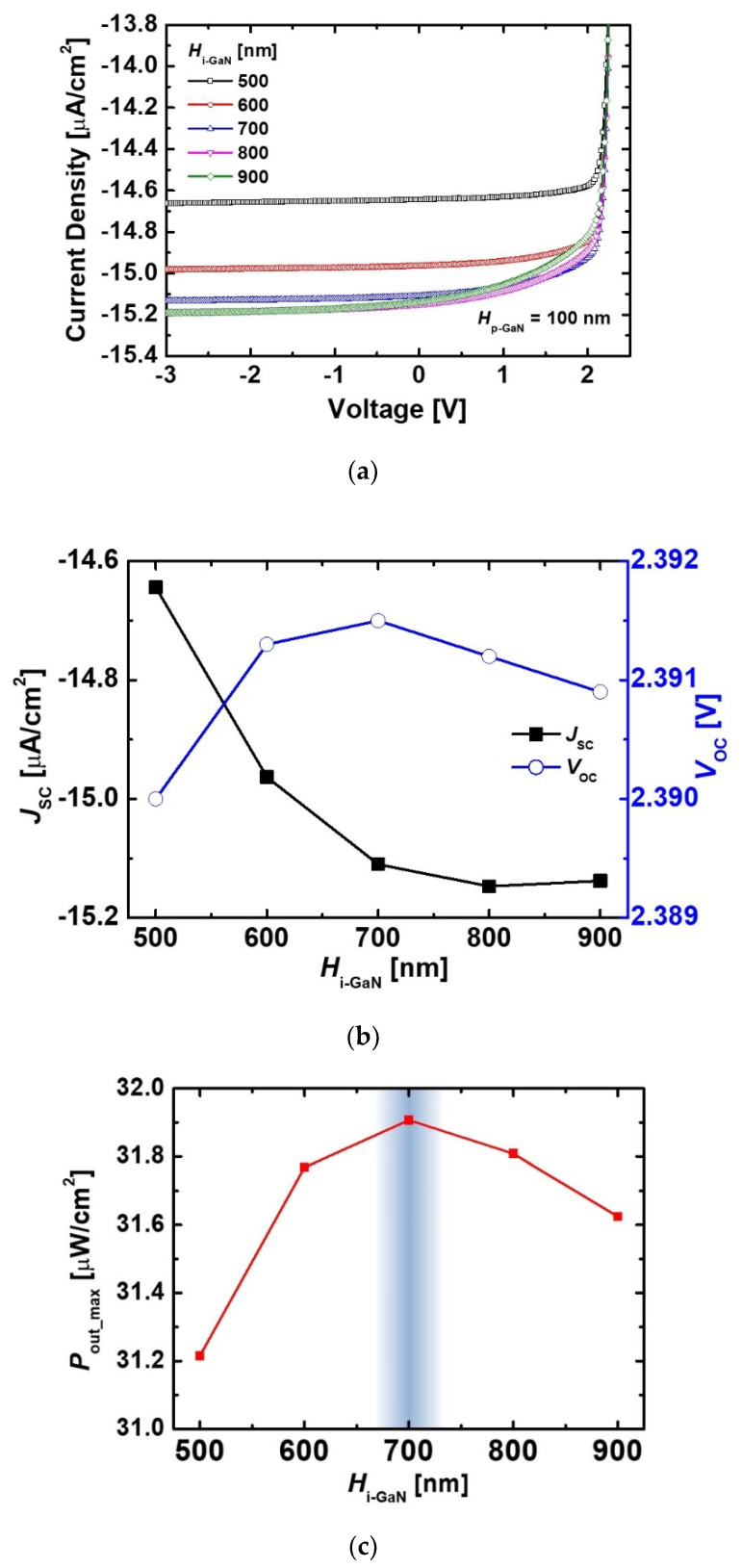
(**a**) Reverse current characteristics of the irradiated diodes with different *H*_i-GaN_ values. (**b**) Variations of *J*_SC_, *V*_OC_ and (**c**) *P*_out_max_ as a function of *H*_i-GaN_. The *H*_p-GaN_ was fixed as 100 nm. The *D*_p-GaN_, *D*_i-GaN_, and *D*_n-GaN_ were 5 × 10^17^ cm^−3^, 1 × 10^16^ cm^−3^_,_ and 5 × 10^18^ cm^−3^, respectively.

**Figure 4 micromachines-11-01100-f004:**
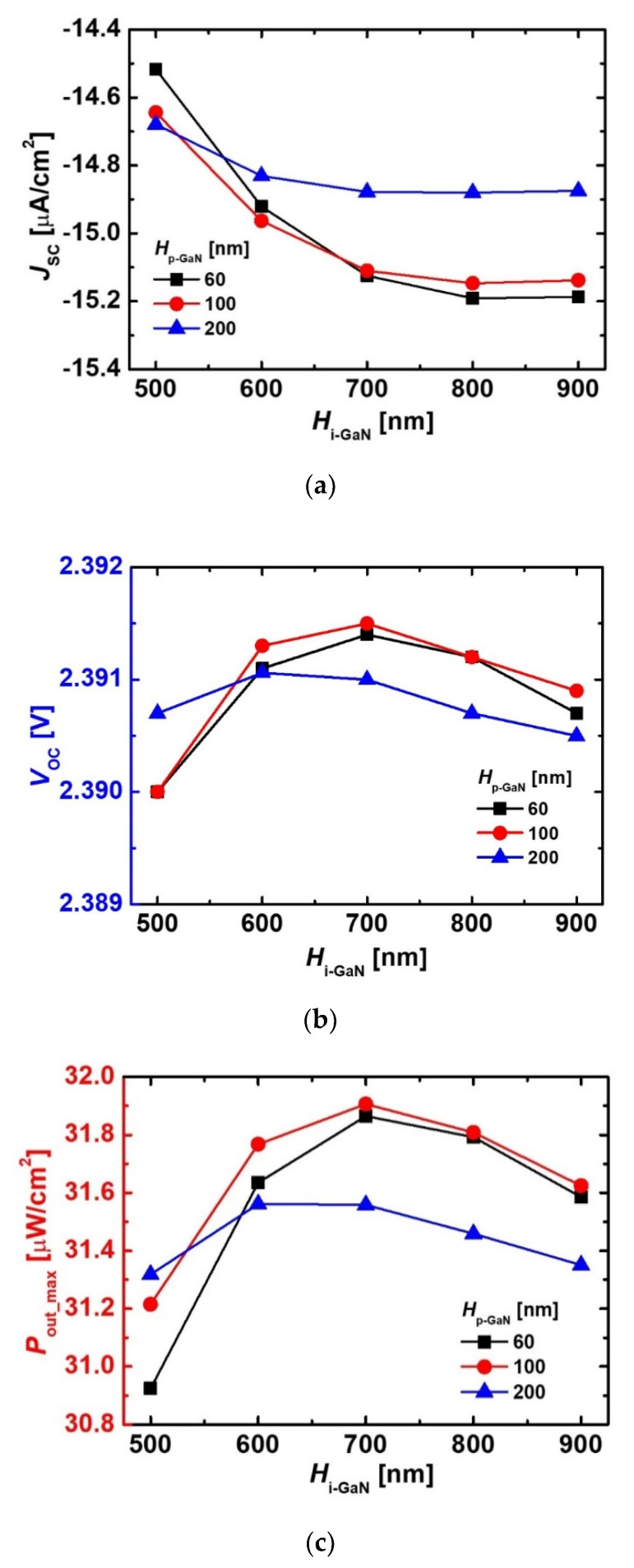
Variations of (**a**) *J*_SC_, (**b**) *V*_OC_ and (**c**) *P*_out_max_ of the irradiated diodes dependent on the *H*_i-GaN_ and *H*_p-GaN_. The *D*_p-GaN_, *D*_i-GaN_, and *D*_n-GaN_ were 5 × 10^17^ cm^−3^, 1 × 10^16^ cm^−3^_,_ and 5 × 10^18^ cm^−3^, respectively.

**Figure 5 micromachines-11-01100-f005:**
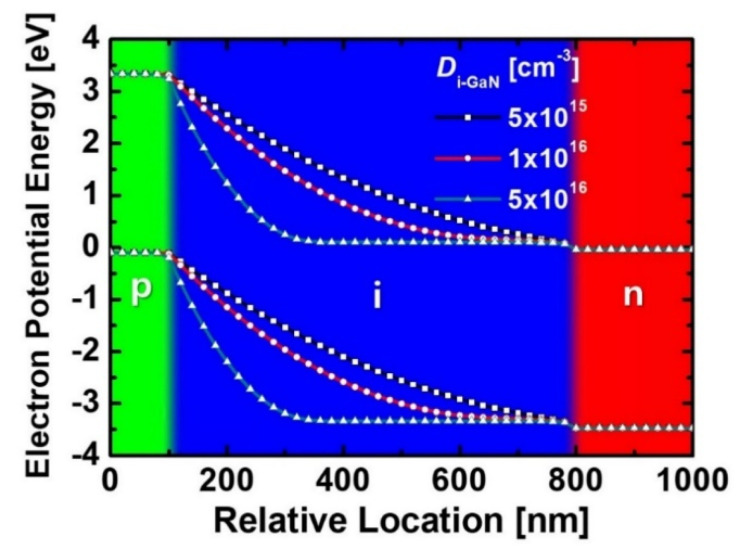
Energy band diagrams of the diodes with different values of *D*_i-GaN_. All the devices were designed with a *H*_p-GaN_ of 100 nm and a *H*_i-GaN_ of 700 nm. The *D*_p-GaN_ and *D*_n-GaN_ of the devices were 5 × 10^17^ cm^−3^ and 5 × 10^18^ cm^−3^, respectively.

**Figure 6 micromachines-11-01100-f006:**
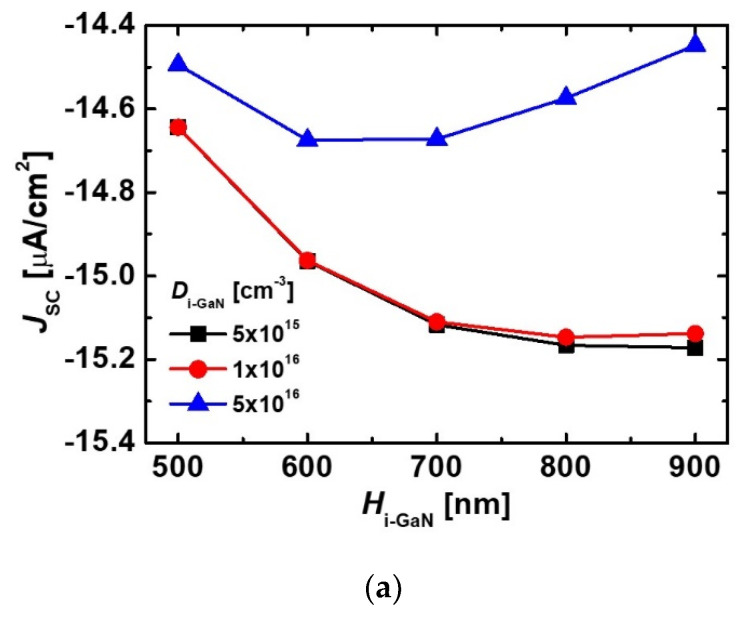
Variations of (**a**) *J*_SC_, (**b**) *V*_OC_ and (**c**) *P*_out_max_ of the e-beam-irradiated diodes depending on *H*_i-GaN_ and *D*_i-GaN_. All devices were designed with a *H*_p-GaN_ of 100 nm and *H*_i-GaN_ of 700 nm. The *D*_p-GaN_ and *D*_n-GaN_ of the devices were 5 × 10^17^ cm^−3^ and 5 × 10^18^ cm^−3^, respectively.

**Figure 7 micromachines-11-01100-f007:**
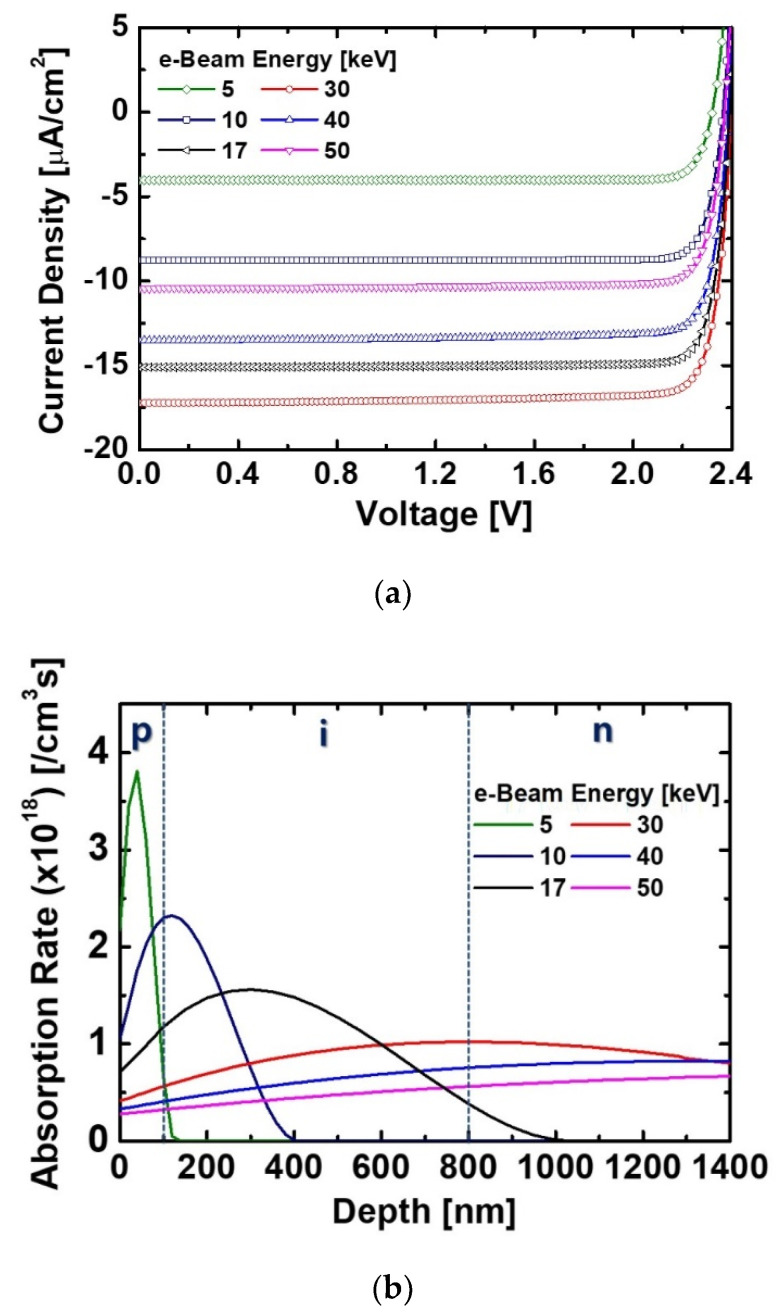
(**a**) Current characteristics of the diodes irradiated by different e-beam energies and (**b**) absorption rate varied by different e-beam energies. All devices were designed with a *H*_p-GaN_ of 100 nm and *H*_i-GaN_ of 700 nm. The *D*_p-GaN_ and *D*_n-GaN_ of the devices were 5 × 10^17^ cm^−3^ and 5 × 10^18^ cm^−3^, respectively.

**Figure 8 micromachines-11-01100-f008:**
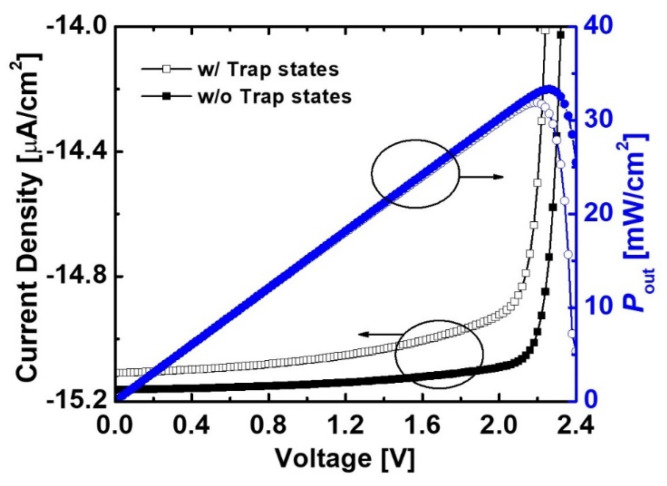
Current characteristics and *P*_out_ of the e-beam-irradiated diodes without and with native trap states. The *H*_p-GaN_ and *H*_i-GaN_ in the diode were 100 nm and 700 nm, respectively. *D*_p-GaN_, *D*_i-GaN_, and *D*_n-GaN_ were 5 *×* 10^17^ cm^−3^, 1 × 10^16^ cm^−3^, and 5 × 10^18^ cm^−3^, respectively.

**Figure 9 micromachines-11-01100-f009:**
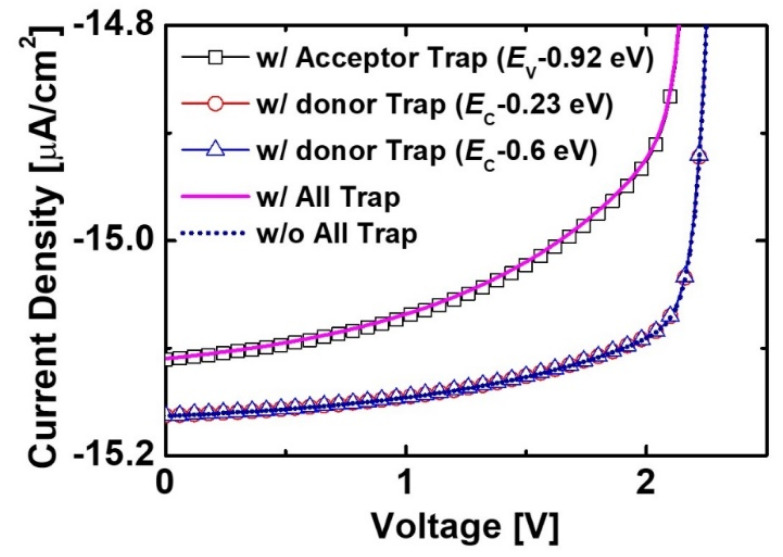
Effects of trap states on current characteristics of the e-beam-irradiated diodes. The *H*_p-GaN_ and *H*_i-GaN_ in the diode were 100 nm and 700 nm, respectively. *D*_p-GaN_, *D*_i-GaN_, and *D*_n-GaN_ were 5 × 10^17^ cm^−3^, 1 × 10^16^ cm^−3^, and 5 × 10^18^ cm^−3^, respectively.

**Figure 10 micromachines-11-01100-f010:**
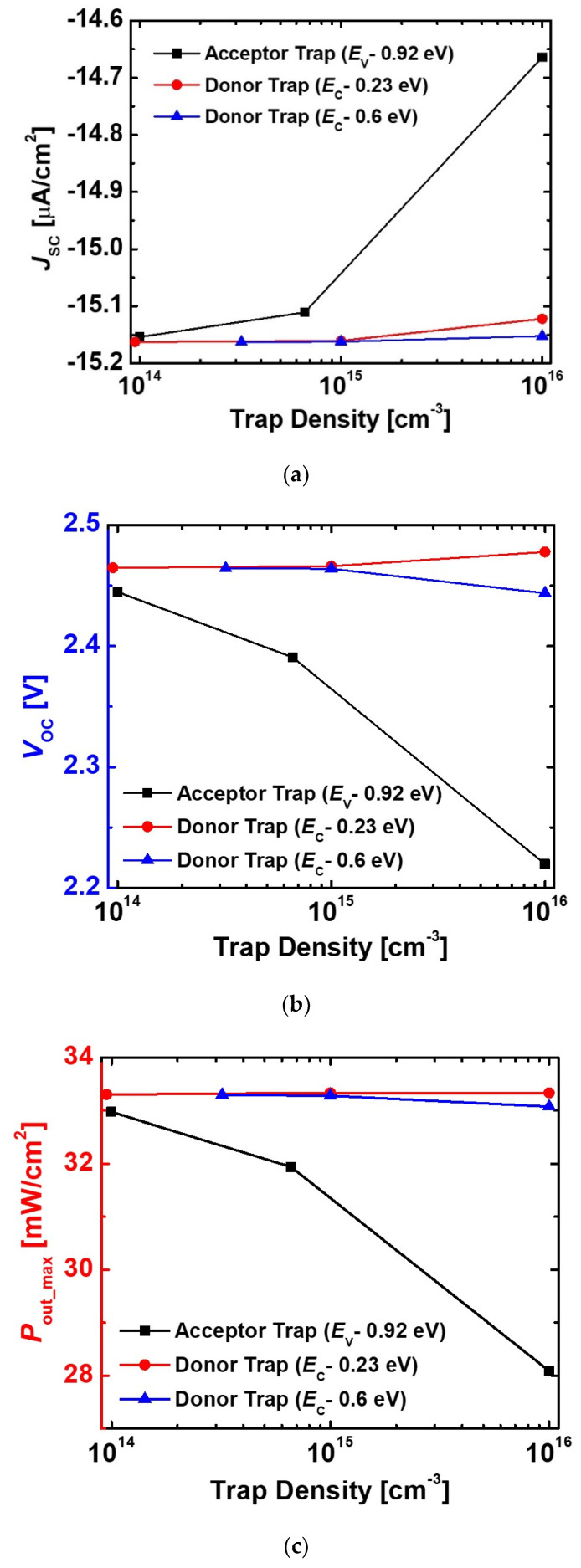
Effects of each donor and acceptor trap states on (**a**) *J*_SC_, (**b**) *V*_OC_, and (**c**) *P*_out_max_ of the e-beam-irradiated diodes. The *H*_p-GaN_ and *H*_i-GaN_ in the diode were 100 nm and 700 nm, respectively. *D*_p-GaN_, *D*_i-GaN_, and *D*_n-GaN_ were 5 × 10^17^ cm^−3^, 1 × 10^16^ cm^−3^, and 5 × 10^18^ cm^−3^, respectively.

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
