# Peer review of "Design and Analysis of Gallium Nitride-Based p-i-n Diode Structure for Betavoltaic Cell with Enhanced Output Power Density"

_micromachines, 2020, doi:10.3390/mi11121100_

Round 1

Reviewer 1 Report

This is an interesting paper, addressing optimization of a GaN betavoltaic. It's primary value is in variation of the geometry of the device. I've read many papers purporting to optimize a betavoltaic, but, inexplicably, many of the geometric parameters are typically not part of the optimization. This paper goes well beyond others I've read in this aspect.

I believe the paper would be improved if two items were fixed:

  1. The grammar needs significant work
  2. The authors have chosen to simulate the device using a monoenergetic electron source of 17 keV. I understand the motivation, given that this is the average energy of the beta particles emitted by a Ni-63 source, but it oversimplifies the true situation. Given that the actual beta particles will be emitted with a spectrum featuring a peak energy of 66 keV, the ranges of the actual particles will vary quite a bit relative to the range of the average particle. As layer depth is a key geometric parameter varied in the optimization, I would expect a dramatic effect of using the spectrum as compared to using the average energy.

Author Response

Reviewer(s)' Comments to Author:
Comments to the Author
This is an interesting paper, addressing optimization of a GaN betavoltaic. It's primary value is in variation of the geometry of the device. I've read many papers purporting to optimize a betavoltaic, but, inexplicably, many of the geometric parameters are typically not part of the optimization. This paper goes well beyond others I've read in this aspect.
I believe the paper would be improved if two items were fixed:

1) The grammar needs significant work

Answer)
Thank you for your comments. We performed English correction in the manuscript.

2) The authors have chosen to simulate the device using a monoenergetic electron source of 17 keV. I understand the motivation, given that this is the average energy of the beta particles emitted by a Ni-63 source, but it oversimplifies the true situation. Given that the actual beta particles will be emitted with a spectrum featuring a peak energy of 66 keV, the ranges of the actual particles will vary quite a bit relative to the range of the average particle. As layer depth is a key geometric parameter varied in the optimization, I would expect a dramatic effect of using the spectrum as compared to using the average energy.

Answer)
We added the figure and sentences for current characteristic and absorption rate of the diodes irradiated by different e-beam energy considering energy spectrum of 63Ni source. We did not compare between performances using the spectrum and using the average energy due to the simulation environment. For accurate evaluation of the performance using energy spectrum of 63Ni source, the probability must be involved in simulation. We are going to examine the evaluation of the performance using energy spectrum of 63Ni source as future work.

-> “The energy spectrum of the 63Ni source exhibited a wide range to a 66 keV peak energy [19]. We additionally confirmed the performances of the optimized diode depending on the injected electron energy. The current characteristics of the diodes irradiated by different e-beam energies is shown in Fig. 7(a). As the energy increased up to 30 keV, the current became higher. This was because many EHPs were generated by a wide distribution of absorption rate as shown in Fig. 7(b). However, when the electrons with an energy above 40 keV were injected, the current of the irradiated device was lower than that of the device irradiated by the 17 keV e-beam. These results revealed that the variations of the current of the irradiated diodes depending on the energy of the e-beam were large. The probability of beta particles generated from the 63Ni source showed a high distribution below 20 keV [19]. Also, the depletion width formed in the i-GaN region was small at about 600 nm (in case of Di-GaN = 1×1016 cm-3), and the diffusion length of GaN can be shortened by native defects. Therefore, in order to achieve a BV cell with high efficiency, it is reasonable to design GaN-based diodes
considering the average energy of the 63Ni source.”
(176th line, page 9 – 188th line, page 10)

Reviewer 2 Report

The basic idea of this paper is fine, and there is value in simulating devices to help identify the limiting factors in device performance, even when not supported by experimental data as in this case.

There are a lot of mistakes in the English throughout the paper, which in places negatively affects the readability.

To start with, the title does not make grammatical sense. It would make more sense as "Design and Analysis of Gallium Nitride-based p-i-n Diode Structure for a Betavoltaic Cell with Enhanced Output Power Density".

The abstract refers to Di-GaN and Hi-GaN without defining these. Or maybe Di-GaN is defined, but this is confused by the word "in" before it. This in an example of where the English makes it hard to understand.

Abstract: Giving a quantitative value of Pout here is meaningless without also defining the electron beam current density.

Section 2: Why were the various parameters assigned the values they were? If these are related to real-world experimental values, then some references would be useful here. For example, some explanation of the doping values for the "intrinsic" GaN would be helpful. The GaN is not intrinsic if doped - is the doping referred to meant to simulate the effect of residual native n-type doping in the i-GaN? If so, where do the values come from? Also, I assume the range of thickness values was informed by the penetration depth of 17 keV electrons in the material, but these values are just stated without any reference to this. And are the contact resistance values taken from the literature or just guessed?

Section 3: As in the abstract, there is no mention of the electron beam current or current density. This makes the quoting of absolute J_SC or P_out values entirely meaningless! Does it even make sense to be using these values as figures of merit, or would it be better to use quantities which were normalized to the rate of incoming electrons (for example, quantum efficiency)? I am not sure, but I know that even if the values used were useful for looking at the *relative* performance of the device during optimization of the doping/thicknesses, they are of no use for quoting as absolute values without further context.

The conclusion is better, in that it refers only to the relative reduction in P_out caused by traps, rather than the absolute values.

The electron irradiation current density will be important not only to put the J_SC and P_out values in context, but also because the relative performance of the device will change as the carrier injection density changes. When the authors quote the missing value, they should also discuss the whether this has been chosen to match a likely real-world scenario in a 63Ni betavoltaic cell, and show calculations or cite references to support this.

Author Response

Comments to the Author
The basic idea of this paper is fine, and there is value in simulating devices to help identify the limiting factors in device performance, even when not supported by experimental data as in this case.

1) There are a lot of mistakes in the English throughout the paper, which in places negatively affects the readability. To start with, the title does not make grammatical sense. It would make more sense as "Design and Analysis of Gallium Nitride-based p-i-n Diode Structure for a Betavoltaic Cell with Enhanced Output Power Density".

Answer)
Thank you for your comments. We performed English correction in the manuscript. Also, We revised the title considering the reviewer comment.

-> The revised title : “Design and Analysis of Gallium Nitride-based p-i-n Diode Structure for Betavoltaic Cell with Enhanced Output Power Density”

2) The abstract refers to Di-GaN and Hi-GaN without defining these. Or maybe Di-GaN is defined, but this is confused by the word "in" before it. This in an example of where the English makes it hard to understand.

Answer)
Thank you for your comments. We added the sentence in the abstract section.

-> “The short-circuit current density (JSC) and open-circuit voltage (VOC) of the 17 keV electron-beam (e-beam)-irradiated diode were evaluated with the variations of design parameters, such as the height and doping concentration of the intrinsic GaN region (Hi-GaN and Di-GaN), which influenced the depletion width in the i-GaN region.”
(14th line – 17th line, page 1)

3) Abstract: Giving a quantitative value of Pout here is meaningless without also defining the electron beam current density.

Answer)
Because we compared Pout of the diodes with the variations of geometrical parameters, we agreed to the reviewer opinion. Thus, we removed a quantitative value of Pout in the abstract section.

4) Section 2: Why were the various parameters assigned the values they were? If these are related to real-world experimental values, then some references would be useful here. For example, some explanation of the doping values for the "intrinsic" GaN would be helpful. The GaN is not intrinsic if doped - is the doping referred to meant to simulate the effect of residual native n-type doping in the i-GaN? If so, where do the values come from?

Answer)
We added the sentence and reference for the doping concentration of the intrinsic GaN.

-> “Here, i-GaN denotes undoped GaN, which is almost an n-type due to the residual donor [21]. The background impurity concentration in undoped GaN grown by metal-organic chemical-vapor deposition (MOCVD) is typically in the range of 1015 to 1017 cm-3, depending on the condition of the reactor [22]. Furthermore, it is difficult to grow undoped GaN with simultaneous low doping concentration and high quality [23]. Thus, the Di-GaN was varied in the range of 5×1015 cm-3 to 5×1016 cm-3 for optimizing the Di-GaN.”
(61th line – 66th line, page 1)

[21] Boguslawski, P.; Briggs, E. L.; Bernholc, J., Native defects in gallium nitride. Phys. Rev. B 1995, 51, 17255-17259.
[22] Alugubelli, S. R.; Fu, H.; Fu, K.; Liu, H.; Zhao, Y.; Ponce, F. A., Dopant profiling in p-i-n GaN
structures using secondary electrons. J. Appl. Phys. 2019, 126, 015704.
[23] Cheng, Z. J.; San, H. S.; Feng, B.; Liu, B.; Chen, X. Y., High open-circuit voltage betavoltaic cell
based on GaN pin homojunction. Electron. Lett. 2011, 47, 720-722.
(page 14)

5) Also, I assume the range of thickness values was informed by the penetration depth of 17 keV electrons in the material, but these values are just stated without any reference to this.

Answer)
We added the sentence and reference for the reason determining the range of thickness values.
->“The ranges of Hp-GaN and Hi-GaN values were determined by considering the penetration depth of 17 keV electrons at about 1 μm [24]. The energy of the e-beam is the average energy of 63Ni [25,26].”
(69th line– 71th line, page 2)

[24] Aydin. S; Kam, E., Investigation of nickel-63 radioisotope-powered GaN betavoltaic nuclear battery. Int. J. Energy Res. 2019, 43, 8725-8738.
(page 14)

6) And are the contact resistance values taken from the literature or just guessed?

Answer)
In the simulation, we applied the acceptable contact resistance values taken from the literature. And, we added the references.
-> “The contact resistance for p-GaN and n-GaN in the devices was 1x10-4 Ω·cm2
[27, 28].”
(71th line– 72th line, page 2)

[27] Wang, D.-F.; Shiwei, F.; Lu, C.; Motayed, A.; Jah, M.; Mohammad, S. N.; Jones, K. A.; Salamanca￾Riba L., Low-resistance Ti/Al/Ti/Au multilayer ohmic contact to n-GaN. J. Appl. Phys. 2001, 89, 6214-6217.

[28] Ho, J.-K.; Jong, C.-S.; Chiu, C. C.; Huang, C.-N.; Chen, C.-Y.; Shih, K.-K., Low-resistance ohmic contacts to p-type GaN. Appl. Phys. Lett. 1999, 74, 1275-1277.
(page 14)

7) Section 3: As in the abstract, there is no mention of the electron beam current or current density. This makes the quoting of absolute J_SC or P_out values entirely meaningless! Does it even make sense to be using these values as figures of merit, or would it be better to use quantities which were normalized to the rate of incoming electrons (for example, quantum efficiency)? I am not sure, but I know that even if the values used were useful for looking at the *relative* performance of the device during optimization of the doping/thicknesses, they are of no use for quoting as absolute values without further context.
The conclusion is better, in that it refers only to the relative reduction in P_out caused by traps, rather than the absolute values. The electron irradiation current density will be important not only to put the J_SC and P_out values in context, but also because the relative performance of the device will change as the carrier injection density changes. When the authors quote the missing value, they should also discuss the whether this has been chosen to match a likely real-world scenario in a 63Ni betavoltaic cell, and show calculations or cite references to support this.

Answer)
Thank you for your kind comments. We compared JSC and Pout of the diodes with the variations of geometrical parameters in the e-beam condition with same electron current. We agreed to the reviewer opinion. Thus, We removed quantitative values of JSC and Pout and compared relatively performances in the result and discussion section. We are going to carry out the simulation in conditions considering a likely real-world scenario in a 63Ni betavoltaic cell as future work.

Round 2

Reviewer 1 Report

The authors attempted to argue that their use of the average energy of the Ni-63 beta particles, rather than the full energy spectrum, is appropriate, but I don't think they make a convincing case. They added results for different beta energies, which is interesting, but then they state: "Therefore, in order to achieve a BV cell with high efficiency, it is reasonable to design GaN-based diodes 187 considering the average energy of the 63Ni source."

I just don't think they can make this strong statement. What they've done is presented an optimized betavoltaic for a monoenergetic beta source. This says nothing about the optimum design for a true Ni-63 source. I'm fine if they must leave consideration of the full spectrum for future work, due to limitations of their simulations, but they should not make conclusions that their data does not support.

Author Response

Comments to the Author
The authors attempted to argue that their use of the average energy of the Ni-63 beta particles, rather than the full energy spectrum, is appropriate, but I don't think they make a convincing case. They added results for different beta energies, which is interesting, but then they state: "Therefore, in order to achieve a BV cell with high efficiency, it is reasonable to design GaN-based diodes 187 considering the average energy of the 63Ni source." I just don't think they can make this strong statement. What they've done is presented an optimized betavoltaic for a monoenergetic beta source. This says nothing about the optimum design for a true Ni-63 source. I'm fine if they must leave consideration of the full spectrum for future work, due to limitations of their simulations, but they should not make conclusions that their data does not support.
Answer)
Thank you for your comments. We revised the sentence in the manuscript.
"Therefore, in order to achieve a BV cell with high efficiency, it is reasonable to design GaN-based diodes considering the average energy of the 63Ni source."
-> “Therefore, in order to achieve a high efficiency BV cell using the 63Ni source, it is necessary to analyze the performances of GaN-based diodes considering the spectrum of the 63Ni source.”
(186th line, page 9– 188th line, page 10)

This manuscript is a resubmission of an earlier submission. The following is a list of the peer review reports and author responses from that submission.